# Designed Measurements for Vector Count Data

[1]**Liming Wang,** [1]**David Carlson,** [2]**Miguel Dias Rodrigues,** [3]**David Wilcox,**
[1]**Robert Calderbank and** [1]**Lawrence Carin**
[1]Department of Electrical and Computer Engineering, Duke University
[2]Department of Electronic and Electrical Engineering, University College London
[3]Department of Chemistry, Purdue University
{liming.w, david.carlson, robert.calderbank, lcarin}@duke.edu
m.rodrigues@ucl.ac.uk     wilcoxds@purdue.edu

## Abstract

We consider design of linear projection measurements for a vector Poisson signal model. The projections are performed on the vector Poisson rate, $X \in \mathbb{R}_+^n$, and the observed data are a vector of counts, $Y \in \mathbb{Z}_+^m$. The projection matrix is designed by maximizing mutual information between $Y$ and $X$, $I(Y;X)$. When there is a latent class label $C \in \{1,\ldots,L\}$ associated with $X$, we consider the mutual information with respect to $Y$ and $C$, $I(Y;C)$. New analytic expressions for the gradient of $I(Y;X)$ and $I(Y;C)$ are presented, with gradient performed with respect to the measurement matrix. Connections are made to the more widely studied Gaussian measurement model. Example results are presented for compressive topic modeling of a document corpora (word counting), and hyperspectral compressive sensing for chemical classification (photon counting).

## 1  Introduction

There is increasing interest in exploring connections between information and estimation theory. For example, mutual information and conditional mean estimation have been discovered to possess close interrelationships. The derivative of mutual information in a scalar Gaussian channel [11] has been expressed in terms of the minimum mean-squared error (MMSE). The connections have also been extended from the scalar Gaussian to the scalar Poisson channel model [12]. The gradient of mutual information in a vector Gaussian channel [17] has been expressed in terms of the MMSE matrix. It has also been found that the relative entropy can be represented in terms of the mismatched MMSE estimates [23, 24]. Recently, parallel results for scalar binomial and negative binomial channels have been established [22, 10].

Inspired by the Lipster-Shiryaev formula [16], it has been demonstrated that for certain channels (or measurement models), investigation of the gradient of mutual information can often lead to a relatively simple formulation, relative to computing mutual information itself. Further, it has been shown that the derivative of mutual information with respect to key system parameters also relates to the conditional mean estimates in other channel settings beyond Gaussian and Poisson models [18].

This paper pursues this overarching theme for a *vector* Poisson measurement model. Results for *scalar* Poisson signal models have been developed recently [12, 1] for signal recovery; the vector results presented here are new, with known scalar results recovered as a special case. Further, we consider the gradient of mutual information for Poisson data in the context of classification, for which there are no previous results, even in the scalar case.

The results we present for optimizing mutual information in vector Poisson measurement models are general, and may be applied to optical communication systems [15, 13]. The specific applications that motivate this study are compressive measurements for vector Poisson data. Direct observation of long vectors of counts may be computationally or experimentally expensive, and therefore it is of interest to design compressive Poisson measurements. Almost all existing results for compres-

sive sensing (CS) directly or implicitly assume a Gaussian measurement model [6], and extension to Poisson measurements represents an important contribution of this paper. To the authors knowledge, the only previous examination of CS with Poisson data was considered in [20], and that paper considered a single special (random) measurement matrix, it did not consider design of measurement matrices, and the classification problems was not addressed. It has been demonstrated *in the context of Gaussian measurements* that designed measurement matrices, using information-theoretic metrics, may yield substantially improved performance relative to randomly constituted measurement matrices [7, 8, 21]. In this paper we extend these ideas to vector Poisson measurement systems, for both signal recovery and classification, and make connections to the Gaussian measurement model. The theory is demonstrated by considering compressive topic modeling of a document corpora, and chemical classification with a compressive photon-counting hyperspectral camera [25].

## 2 Mutual Information for Designed Compressive Measurements

### 2.1 Motivation

A source random variable $X \in \mathbb{R}^n$, with probability density function $P_X(X)$, is sent through a measurement channel, the output of which is characterized by random variable $Y \in \mathbb{R}^m$, with conditional probability density function $P_{Y|X}(Y|X)$; we are interested in the case $m < n$, relevant for compressive measurements, although the theory is general. Concerning $P_{Y|X}(Y|X)$, in this paper we focus on Poisson measurement models, but we also make connections to the much more widely considered Gaussian case. For the Poisson and Gaussian measurement models the *mean* of $P_{Y|X}(Y|X)$ is $\Phi X$, where $\Phi \in \mathbb{R}^{m \times n}$ is the measurement matrix. For the Poisson case the mean may be modified as $\Phi X + \lambda$ for "dark current" $\lambda \in \mathbb{R}_+^m$, and positivity constraints are imposed on the elements of $\Phi$ and $X$.

Often the source statistics are characterized as a mixture model: $P_X(X) = \sum_{c=1}^{L} \pi_c P_{X|C}(X|C = c)$, where $\pi_c > 0$ and $\sum_{c=1}^{L} \pi_c = 1$, and $C$ may correspond to a latent class label. In this context, for each draw $X$ there is a latent class random variable $C \in \{1, \dots, L\}$, where the probability of class $c$ is $\pi_c$.

Our goal is to design $\Phi$ such that the observed $Y$ is most informative about the underlying $X$ or $C$. When the interest is in recovering $X$, we design $\Phi$ with the goal of maximizing mutual information $I(X;Y)$, while when interested in inferring $C$ we design $\Phi$ with the goal of maximizing $I(C;Y)$.

To motivate use of the mutual information as the design metric, we note several results from the literature. For the case in which we are interested in recovering $X$ from $Y$, it has been shown [19] that

$$\text{MMSE} \geq \frac{1}{2\pi e} \exp\{2[h(X) - I(X;Y)]\} \tag{1}$$

where $h(X)$ is the differential entropy of $X$ and $\text{MMSE} = \mathbb{E}\{\text{trace}[(X - \mathbb{E}(X|Y))(X - \mathbb{E}(X|Y))^T]\}$ is the minimum mean-square error.

For the classification problem, we define the Bayesian classification error as $P_e = \int P_Y(y)[1 - \max_c P_{C|Y}(c|y)]dy$. It has been shown in [14] that

$$[H(C|Y) - H(P_e)]/\log L \leq P_e \leq \frac{1}{2} H(C|Y) \tag{2}$$

where $H(C|Y) = H(C) - I(C;Y)$, $0 \leq H(P_e) \leq 1$, and $H(\cdot)$ denotes the entropy of a discrete random variable. By minimizing $H(C|Y)$ we minimize the upper bound to $P_e$, and since $H(C)$ is independent of $\Phi$, to minimize the upper bound to $P_e$ our goal is to design $\Phi$ such that $I(C;Y)$ is maximized.

### 2.2 Existing results for Gaussian measurements

There are recent results for the gradient of mutual information for vector Gaussian measurements, which we summarize here. Consider the case $C \sim P_C(C)$, $X|C \sim P_{X|C}(X|C)$, and $Y|X \sim \mathcal{N}(Y; \Phi X, \Lambda^{-1})$, where $\Lambda \in \mathbb{R}^{m \times m}$ is a known precision matrix. Note that $P_C$ and $P_{X|C}$ are arbitrary, while $P_{Y|X} = \mathcal{N}(Y; \Phi X, \Lambda^{-1})$ corresponds to a Gaussian measurement with mean $\Phi X$.

It has been established that the gradient of mutual information between the input and the output of the vector Gaussian channel model obeys [17]

$$\nabla_\Phi I(X;Y) = \Lambda \Phi E, \tag{3}$$

where $E = \mathbb{E}\left[(X - \mathbb{E}(X|Y))(X - \mathbb{E}(X|Y))^T\right]$ denotes the MMSE matrix. The gradient of mutual information between the class label and the output for the vector Gaussian channel is [8]

$$\nabla_\Phi I(C;Y) = \Lambda \Phi \tilde{E}, \tag{4}$$

where $\tilde{E} = \mathbb{E}\left[(\mathbb{E}(X|Y,C) - \mathbb{E}(X|Y))(\mathbb{E}(X|Y,C) - \mathbb{E}(X|Y))^T\right]$ denotes the equivalent MMSE matrix.

## 2.3 Conditional-mean estimation

Note from the above discussion that for a Gaussian measurement, $\nabla_\Phi I(X;Y) = \mathbb{E}[f(X, \mathbb{E}(X|Y))]$ and $\nabla_\Phi I(C;Y) = \mathbb{E}[g(\mathbb{E}(X|Y,C), \mathbb{E}(X|Y))]$, where $f(\cdot)$ and $g(\cdot)$ are matrix-valued functions of the respective arguments. These results highlight the connection between the gradient of mutual information with respect to the measurement matrix $\Phi$ and conditional-mean estimation, constituted by $\mathbb{E}(X|Y)$ and $\mathbb{E}(X|Y,C)$. We will see below that these relationships hold as well for the vector Poisson case, with *distinct* functions $\tilde{f}(\cdot)$ and $\tilde{g}(\cdot)$.

# 3 Vector Poisson Data

## 3.1 Model

The vector Poisson channel model is defined as

$$\text{Pois}(Y; \Phi X + \lambda) = P_{Y|X}(Y|X) = \prod_{i=1}^{m} P_{Y_i|X}(Y_i|X) = \prod_{i=1}^{m} \text{Pois}(Y_i; (\Phi X)_i + \lambda_i) \tag{5}$$

where the random vector $X = (X_1, X_2, \ldots, X_n) \in \mathbb{R}_+^n$ represents the channel input, the random vector $Y = (Y_1, Y_2, \ldots, Y_m) \in \mathbb{Z}_+^m$ represents the channel output, $\Phi \in \mathbb{R}_+^{m \times n}$ represents a measurement matrix, and the vector $\lambda = (\lambda_1, \lambda_2, \ldots, \lambda_m) \in \mathbb{R}_+^m$ represents the dark current.

The vector Poisson channel model associated with arbitrary $m$ and $n$ is a generalization of the scalar Poisson model, for which $m = n = 1$ [12, 1]. In the scalar case $P_{Y|X}(Y|X) = \text{Pois}(Y; \phi X + \lambda)$, where here *scalar* random variables $X \in \mathbb{R}_+$ and $Y \in \mathbb{Z}_+$ are associated with the input and output of the scalar channel, respectively, $\phi \in \mathbb{R}_+$ is a scaling factor, and $\lambda \in \mathbb{R}_+$ is associated with the dark current.

The goal is to design $\Phi$ to maximize the mutual information between $X$ and $Y$. Toward that end, we consider the gradient of mutual information with respect to $\Phi$: $\nabla_\Phi I(X;Y) = [\nabla_\Phi I(X;Y)_{ij}]$, where $\nabla_\Phi I(X;Y)_{ij}$ represents the $(i,j)$-th entry of the matrix $\nabla_\Phi I(X;Y)$. We also consider the gradient with respect to the vector dark current, $\nabla_\lambda I(X;Y) = [\nabla_\lambda I(X;Y)_i]$, where $\nabla_\lambda I(X;Y)_i$ represents the $i$-th entry of the vector $\nabla_\lambda I(X;Y)$. For a mixture-model source $P_X(X) = \sum_{c=1}^{L} \pi_c P_{X|C=c}(X|C=c)$, for which there is more interest in recovering $C$ than in recovering $X$, we seek $\nabla_\Phi I(C;Y)$ and $\nabla_\lambda I(C;Y)$.

## 3.2 Gradient of Mutual Information for Signal Recovery

In order to take full generality of the input distribution into consideration, we utilize the Radon-Nikodym derivatives to represent the probability measures of interests. Consider random variables $X \in \mathbb{R}^n$ and $Y \in \mathbb{R}^m$. Let $f_{Y|X}^\theta$ be the Radon-Nikodym derivative of probability measure $P_{Y|X}^\theta$ with respect to an arbitrary measure $Q_Y$, provided that $P_{Y|X}^\theta$ is absolutely continuous with respect to $Q_Y$, *i.e.*, $P_{Y|X}^\theta \ll Q_Y$. $\theta \in \mathbb{R}$ is a parameter. $f_Y^\theta$ is the Radon-Nikodym derivative of the probability measure $P_Y^\theta$ with respect to $Q_Y$ provided that $P_Y^\theta \ll Q_Y$. Note that in the continuous or discrete case, $f_{Y|X}^\theta$ and $f_Y^\theta$ are simply probability density or mass functions with $Q_Y$ chosen to be the Lebesgue measure or the counting measure, respectively. We note that similar notation is also used for the signal classification case, except that we may also need to condition both on $X$ and $C$. Some results of the paper require the assumption on the regularity conditions (RC), which are listed in the Supplementary Material. We will assume all four regularity conditions RC1–RC4 whenever necessary in the proof and the statement of the results. Recall [9] that for a function $f(x, \theta)$ : $\mathbb{R}^n \times \mathbb{R} \to \mathbb{R}$ with a Lebesgue measure $\mu$ on $\mathbb{R}^n$, we have $\frac{\partial}{\partial \theta} \int f(x, \theta) d\mu(x) = \int \frac{\partial}{\partial \theta} f(x, \theta) d\mu(x)$, if $f(x, \theta) \leq g(x)$, where $g \in L^1(\mu)$. Hence, in light of this criterion, it is straightforward to verify that the RC are valid for many common distributions of $X$. Proofs of the below theorems are provided in the Supplementary Material.

**Theorem 1.** *Consider the vector Poisson channel model in* (5). *The gradient of mutual information between the input and output of the channel, with respect to the matrix $\Phi$, is given by:*

$$[\nabla_\Phi I(X;Y)_{ij}] = \big[\mathbb{E}\left[X_j \log((\Phi X)_i + \lambda_i)\right] - \mathbb{E}\left[\mathbb{E}[X_j|Y] \log \mathbb{E}[(\Phi X)_i + \lambda_i|Y]\right]\big], \quad (6)$$

*and with respect to the dark current is given by:*

$$[\nabla_\lambda I(X;Y)_i] = \big[\mathbb{E}[\log((\Phi X)_i + \lambda_i)] - \mathbb{E}[\log \mathbb{E}[(\Phi X)_i + \lambda_i|Y]]\big]. \quad (7)$$

*irrespective of the input distribution $P_X(X)$, provided that the regularity conditions hold.*

### 3.3 Gradient of Mutual Information for Classification

**Theorem 2.** *Consider the vector Poisson channel model in* (5) *and mixture signal model. The gradient with respect to $\Phi$ of mutual information between the class label and output of the channel is*

$$[\nabla_\Phi I(C;Y)_{ij}] = \mathbb{E}\left[\mathbb{E}[X_j|Y,C] \log \frac{\mathbb{E}[(\Phi X)_i + \lambda_i|Y,C]}{\mathbb{E}[(\Phi X)_i + \lambda_i|Y]}\right], \quad (8)$$

*and with respect to the dark current is given by*

$$(\nabla_\lambda I(C;Y))_i = \mathbb{E}\left[\log \frac{\mathbb{E}[(\Phi X)_i + \lambda_i|Y,C]}{\mathbb{E}[(\Phi X)_i + \lambda_i|Y]}\right]. \quad (9)$$

*irrespective of the input distribution $P_{X|C}(X|C)$, provided that the regularity conditions hold.*

### 3.4 Relationship to known scalar results

It is clear that Theorem 1 represents a multi-dimensional generalization of Theorems 1 and 2 in [12]. The scalar result follows immediately from the vector counterpart by taking $m = n = 1$.

**Corollary 1.** *For the scalar Poisson channel model $P_{Y|X}(Y|X) = Pois(Y;\phi X + \lambda)$, we have*

$$\frac{\partial}{\partial \phi} I(X;Y) = \mathbb{E}\left[X \log((\phi X) + \lambda)\right] - \mathbb{E}\left[\mathbb{E}[X|Y] \log \mathbb{E}[\phi X + \lambda|Y]\right], \quad (10)$$

$$\frac{\partial}{\partial \lambda} I(X;Y) = \mathbb{E}[\log(\phi X + \lambda)] - \mathbb{E}[\log \mathbb{E}[\phi X + \lambda|Y]]. \quad (11)$$

*irrespective of the input distribution $P_X(X)$, provided that the regularity conditions hold.*

While the scalar result in [12] for signal recovery is obtained as a special case of our Theorem 1, for recovery of the class label $C$ there are no previous results for our Theorem 2, even in the scalar case.

### 3.5 Conditional mean and generalized Bregman divergence

Considering the results in Theorem 1, and recognizing that $\mathbb{E}[(\Phi X) + \lambda|Y] = \Phi\mathbb{E}(X|Y) + \lambda$, it is clear that for the Poisson case $\nabla_\Phi I(X;Y) = \mathbb{E}[\tilde{f}(X,\mathbb{E}(X|Y))]$. Similarly, for the classification case, $\nabla_\Phi I(C;Y) = \mathbb{E}[\tilde{g}(\mathbb{E}(X|Y,C),\mathbb{E}(X|Y))]$. The gradient with respect to the dark current $\lambda$ has no analog for the Gaussian case, but similarly we have $\nabla_\lambda I(X;Y) = \mathbb{E}[\tilde{f}_1(X,\mathbb{E}(X|Y))]$ and $\nabla_\lambda I(C;Y) = \mathbb{E}[\tilde{g}_1(\mathbb{E}(X|Y,C),\mathbb{E}(X|Y))]$.

For the *scalar* Poisson channel in Corollary 1, it has been shown in [1] that $\frac{\partial}{\partial \phi} I(X;Y) = \mathbb{E}[\ell(X,\mathbb{E}(X|Y))]$, where $\ell(X,\mathbb{E}(X|Y))$ is defined by the right side of (10), and is related to the Bregman divergence [5, 2].

While beyond the scope of this paper, one may show that $\tilde{f}(X,\mathbb{E}(X|Y))$ and $\tilde{g}(\mathbb{E}(X|Y,C),\mathbb{E}(X|Y))$ may be interpreted as *generalized* Bregman divergences, where here the generalization is manifested by the fact that these are matrix-valued measures, rather than the scalar one in [1]. Further, for the vector Gaussian cases one may also show that $f(X,\mathbb{E}(X|Y))$ and $g(\mathbb{E}(X|Y,C),\mathbb{E}(X|Y))$ are also generalized Bregman divergences. These facts are primarily of theoretical interest, as they do not affect the way we perform computations. Nevertheless, these theoretical results, through generalized Bregman divergence, underscore the primacy the conditional mean estimators $\mathbb{E}(X|Y)$ and $\mathbb{E}(X|Y,C)$ within the gradient of mutual information with respect to $\Phi$, for both the Gaussian and Poisson vector measurement models.

# 4 Applications

## 4.1 Topic Models

Consider the case for which the Poisson rate vector for document $d$ may be represented $X_d = \Psi S_d$, where $X_d \in \mathbb{R}_+^n$, $\Psi \in \mathbb{R}_+^{n \times T}$ and $S_d \in \mathbb{R}_+^T$. Here $T$ represents the number of topics, and in the context of documents, $n$ represents the total number of words in dictionary $\mathcal{D}$. The count for the number of times each of the $n$ words is manifested in document $d$ may often be modeled as $Y_d | S_d \sim \text{Pois}(Y_d; \Psi S_d)$; see [26] and the extensive set of references therein.

Rather than counting the number of times each of the $n$ words are separately manifested, we may more efficiently count the number of times words in particular *subsets* of $\mathcal{D}$ are manifested. Specifically, consider a compressive measurement for document $d$, as $Y_d | X_d \sim \text{Pois}(Y_d; \Phi X_d)$, where $\Phi \in \{0,1\}^{m \times n}$, with $m \ll n$. Let $\phi_k \in \{0,1\}^n$ represent the $k$th row of $\Phi$, with $Y_{dk}$ the $k$th component of $Y_d$. Then $Y_{dk} | X_d \sim \text{Pois}(Y_{dk}; \phi_k^T X_d)$ is equal in distribution to $Y_{dk} = \sum_{i=1}^n \tilde{Y}_{dki}$, where $\tilde{Y}_{dki} | X_{di} \sim \text{Pois}(\phi_{ki} X_{di})$, with $\phi_{ki} \in \{0,1\}$ the $i$th component of $\phi_k$ and $X_{di}$ the $i$th component of $X_d$. Therefore, $Y_{dk}$ represents the number of times words in the *set* defined by the non-zero elements of $\phi_k$ are manifested in document $d$; $Y_d$ therefore represents the number of times words are manifested in a document in $m$ distinct sets.

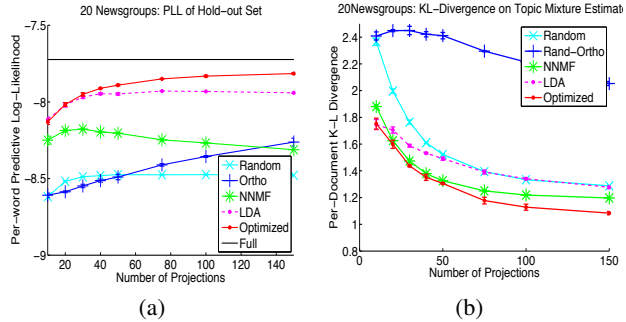

(a)                    (b)

Figure 1: Results on the 20 Newsgroups dataset. Random denotes a random binary matrix with 1% non-zero values. Rand-Ortho denotes a random binary matrix restricted to an orthogonal matrix with one non-zero entry per column. Optimized denotes the methods discussed in Section 4.3. Full denotes when each word is observed. The error estimates were obtained by running the algorithm over 10 different random splits of the corpus. (a) Per-word predictive log-likelihood estimate versus the number of projections. (b) KL Divergence versus the number of projections.

Our goal is to use the theory developed above to design the *binary* $\Phi$ such that the compressive $Y_d | X_d \sim \text{Pois}(Y_d; \Phi X_d)$ is as informative as possible. In our experiments we assume that $\Psi$ may be learned separately based upon a small subset of the corpus, and then with $\Psi$ so fixed the statistics of $X_d$ are driven by the statistics of $S_d$. When performing learning of $\Psi$, each column of $\Psi$ is assumed drawn from an $n$-dimensional Dirichlet distribution, and $S_d$ is assumed drawn from a gamma process, as specified in [26]. We employ variational Bayesian (VB) inference on this model [26] to estimate $\Psi$ (and retain the mean).

With $\Psi$ so fixed, we then design $\Phi$ under two cases. For the case in which we are interested in inferring $S_d$ from the compressive measurements, *i.e.*, based on counts of words in sets, we employ a gamma process prior for $p_S(S_d)$, as in [26]. The result in Theorem 1 is then used to perform gradients for design of $\Phi$. For the classification case, for each document class $c \in \{1, \dots, L\}$ we learn a $p(S_d | C)$ based on a training sub-corpus for class $C$. This is done for all document classes, and we design a compressive matrix $\Phi \in \{0,1\}^{m \times n}$, with gradient performed using Theorem 2.

In the testing phase, using held-out documents, we employ the matrix $\Phi$ to group the counts of words in document $d$ into counts on $m$ sets of words, with sets defined by the rows of $\Phi$. Using these $Y_d$, which we assume are drawn $Y_d | S_d \sim \text{Pois}(Y_d; \Phi \Psi S_d)$, for known $\Phi$ and $\Psi$, we then use VB computations for the model in [26] to infer a posterior distribution on $S_d$ or class $C$, depending on the application. The VB inference for this model was not considered in [26], and the update equations are presented in the Supplementary Material.

## 4.2 Model for Chemical Sensing

The model employed for the chemical sensing [25] considered below is very similar in form to that used for topic modeling, so we reuse notation. Assume that there are $T$ fundamental (building-block) chemicals of interest, and that the hyperspectral sensor performs measurements at $n$ wavelengths. Then the observed data for sample $d$ may be represented $Y_d | S_d \sim \text{Pois}(Y_d; \Psi S_d + \lambda)$, where $Y_d \in \mathbb{Z}_+^n$ represents the count of photons at the $n$ sensor wavelengths, $\lambda \in \mathbb{R}_+^n$ represents the sensor dark current, and the $t$th column of $\Psi \in \mathbb{R}_+^{n \times T}$ reflects the mean Poisson rate for chemical $t$ (the

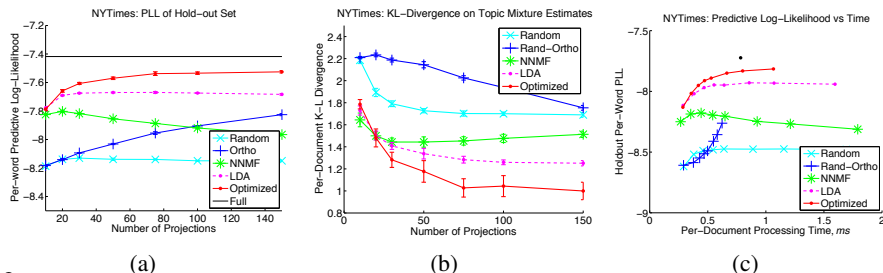

(a)            (b)            (c)

Figure 2: Results on the NYTimes corpus. Optimized denotes the methods discussed in Section 4.3. Full denotes when each word is observed. The error estimates were obtained by running the algorithm over 10 different random subsets of 20,000 documents. (a) Predictive log-likelihood estimate versus the number of projections. (b) KL Divergence versus the number of projections. (c) Predictive log-likelihood versus processing time.

different chemicals play a role analogous to topics). The vector $S_d \in \mathbb{R}_+^T$ reflects the amount of each fundamental chemical present in the sample under test.

For the compressive chemical-sensing system discussed in Section 4.5, the measurement matrix is again binary, $\Phi \in \{0,1\}^{m \times n}$. Through calibrations and known properties of chemicals and characteristics of the camera, one may readily constitute $\Psi$ and $\lambda$, and a model similar to that employed for topic modeling is utilized to model $S_d$; here $\lambda$ is a characteristic of the camera, and is not optimized. In the experiments reported below the analysis of the chemical-sensing data is performed analogously to how the documents were modeled (which we detail), and therefore no further modeling details are provided explicitly for the chemical-sensing application, for brevity. For the chemical sensing application, the goal is to classify the chemical sample under test, and therefore $\Phi$ is defined based on optimization using the Theorem 2 gradient.

### 4.3 Details on Designing $\Phi$

We wish to use Theorems 1 and 2 to design a binary $\Phi$, for the document-analysis and chemical-sensing applications. To do this, instead of directly optimizing $\Phi$, we put a logistic link on each value $\Phi_{ij} = \text{logit}(M_{ij})$. We can state the gradient with respect to $M$ as:

$$[\nabla_M I(X;Y)_{ij}] = [\nabla_\Phi I(X;Y)_{ij}][\nabla_M \Phi_{ij}] \tag{12}$$

Similar results hold for $\nabla_M I(C;Y)_{ij}$. $\Phi$ was initialized at random, and we threshold the logistic at 0.5 to get the final binary $\Phi$.

To estimate the expectations needed for the results in Theorems 1 and 2, we used Monte Carlo integration methods, where we simulated $X$ and $Y$ from the appropriate distribution. The number of samples in the Monte Carlo integration was set to $n$ (data dimension), and 1000 gradient steps were used for optimizing $\Phi$.

The explicit forms for the gradients in Theorems 1 and 2 play an important role in making optimization of $\Phi$ tractable for the practical applications considered here. One could in principle take a brute-force gradient of $I(Y;X)$ and $I(Y;C)$ with respect to $\Phi$, and evaluate all needed integrals via Monte Carlo sampling. This leads to a cumbersome set of terms that need be computed. The "clean" forms of the gradients in Theorems 1 and 2 significantly simplified design implementation within the below experiments, with the added value of allowing connections to be made to the Gaussian measurement model.

### 4.4 Examples for Document Corpora

We demonstrate designed projections on the NYTimes and 20 Newsgroups data. The NYTimes data has $n = 8000$ unique words, and the Newsgroup data has $n = 8052$ unique words. When learning $\Psi$, we placed the prior $\text{Dir}(0.1, \ldots, 0.1)$ on the columns of $\Psi$, and the components $S_{dk}$ had a prior $\text{Gamma}(0.1, 0.1)$. We tried many different settings for these priors, and as in [26], the learned $\Psi$ was insensitive to "reasonable" settings. The number of topics (columns) in $\Psi$ was set to $T = 100$. In addition to designing $\Phi$ using the proposed theory, we also considered four comparative designs: ($i$) binary $\Phi$ constituted uniformly at random, with 1% of the entries non-zero; ($ii$) orthogonal binary rows of $\Phi$, with one non-zero element in each column selected uniformly at random; ($iii$) performing non-negative matrix factorization [3] on (NNMF) $\Psi$, and projecting onto the principal vectors; and ($iv$) performing latent Dirichlet allocation [4] on the documents, and projecting onto the topic-dependent probabilities of words. For ($iii$) and ($iv$), the top (highest amplitude) 5% of

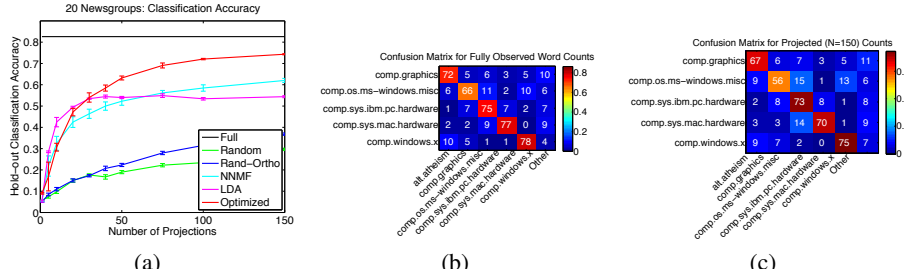

(a)                                    (b)                                    (c)

Figure 3: (a) Classification accuracy of projected measurements and the fully observed case. Random uses 10% non-zero values, Ortho is a random matrix limited to orthogonal projections, and Optimized uses designed projections. The error bars are the standard deviation of the algorithm run independently on 10 random splits of the dataset. (b) Subset of confusion matrix of of the fully observed counts. White numbers denote percentage of documents classified in that manner. Only those classes in the "comp" subgroup are shown. The "comp" group is the *least accurate* subgroup. (c) The confusion matrix on the "comp" subgroup for 150 compressive measurements.

the words in each vector on which we project (*e.g.*, topic) were set to have projection amplitude 1, and all the rest were set to zero. The settings on ($i$), ($iii$) and ($iv$), *i.e.*, with regard to the fraction of words with non-zero values in $\Phi$, were those that yielded the best results (other settings often performed *much* worse).

We show results using two metrics, Kullback-Leibler (KL) divergence and predictive log-likelihood. For the KL divergence, we compare the topic mixture learned from the projection measurements to the topic mixture learned from the case where each word is observed (no compressive measurement). We define the topic mixture $S'_d$ as the normalized version of $S_d$. We calculate $D_{KL}(S'_{d,p}||S'_{d,f}) = \sum_{k=1}^{K} S'_{dk,p} \log(S'_{dk,p}/S'_{dk,f})$, where $S'_{dk,p}$ is the relative weight on document $d$, topic $k$ for the full set of words, and $S'_{dk,p}$ is the same for the compressive topic model. We also calculate per-word predictive log-likelihood. Because different projection metrics are in different dimensions, we use 75% of a document's words to get the projection measurements $Y_d$ and use the remaining 25% as the original word tokens $W_d$. We then calculate the predictive log-likelihood (PLL) as $\log(W_d|\Psi, \Phi, Y_d)$.

We split the 20 Newgroups corpus into 10 random splits of 60% training and 40% testing to get an estimate of uncertainty. The results are shown in Figure 1. Figure 1(a) shows the per-word predictive log-likelihood (PLL). At very low numbers of compressive measurements we get similar PLL between the designed matrix and the random methods. As we increase the number of measurements, we get dramatic improvements by optimizing the sensing matrix and the optimized methods quickly approach the fully observed case. The same trends can be seen in the KL divergence shown in Figure 1(b). Note that the relative quality of the NNMF and LDA based designs of $\Phi$ depends on the metric (KL or PLL), but for both metrics the proposed mutual-information-based design of $\Phi$ yields best performance.

To test the NYTimes corpus, we split the corpus into 10 random subsets with 20,000 training documents and 20,000 testing documents. The results are shown in Figure 2. As in the 20 Newsgroups results, the predictive log-likelihood and KL divergence of the random and designed measurements are similar when the number of projections are low. As we increase the number of projections the optimized projection matrix offers dramatic improvements over the random methods. We also consider predictive log-likelihood versus time in Figure 2(c). The compressive measurements give near the same performance with half the per-document processing time. Since the total processing time increases linearly with the total number of documents, a 50% decrease in processing time can make a significant difference in large corpora.

We also consider the classification problem over the 20 classes in the 20 Newsgroups dataset, split into 10 groups of 60% training and 40% testing. We learn a $\Psi$ with $T = 20$ columns (topics) and with the prior on the columns as above. Within the prior, we draw $S_{dc_d}|c_d \sim \text{Gamma}(1,1)$ and $S_{dc'}|c_d = 0$ for all $c' \neq c_d$. Separate topics are associated with each of the 20 classes, and we use the MAP estimate to get the class label $c_d^* = \arg\max(c|Y_d)$. Classification versus number of projections for random projections and designed projections are shown in Figure 3(a). It is also useful to look at the type of errors made in the classifier when we use the designed projections. Figure 3(b) and Figure 3(c) show the newsgroups under the "comp" (computer) heading, which is the *least*

*accurate* section. In the compressed case, many of the additional errors go into nearby topics with overlapping ideas. For example, most additional misclassifications in "comp.os.ms-windows.misc" go into "comp.sys.ibm.pc.hardware" and "comp.windows.x," which have many similar discussions. Additionally, 4% of the articles were originally posted in more than one topic, showing the intimate relationship between similar discussion groups, and so misclassifying into a related (and overlapping) class is less of a problem than misclassification into a completely disjoint class.

### 4.5 Poisson Compressive Sensing for Chemical Classification

We consider chemical sensing based on the wavelength-dependent signature of chemicals, at optical frequencies (here we consider a 850-1000 nm laser system). In Figure 4(a) the measurement system is summarized; details of this system are described in [25]. In Part 1 of Figure 4(a) multi-wavelength photons are scattered off a chemical sample. In Part 2 of this figure a volume holographic grating (VHG) is employed to diffract the photons in a wavelength-dependent manner, and therefore photons are distributed spatially across a digital mirror microdevice (DMD); distinct wavelengths are associated with each micromirror. The DMD consists of $1920 \times 1080$ aluminum mirrors. Each mirror is in a binary state, either reflecting light back to a detector, or not. Each mirror approximately samples a single wavelength, as a result of the VHG, and the photon counter counts all photons at wavelengths for which the mirrors direct light to the sensor. Hence, the sensor counts all photons at a subset of the wavelengths, those for which the mirror is at the appropriate angle.

The measurement may be represented $Y|S_d \sim \text{Pois}[\Phi(\Psi S_d + \lambda_0)]$, where $\lambda_0 \in \mathbb{R}_+^n$ is known from calibration. The elements of the rate vector of $\lambda_0$ vary from .07 to 1.5 per bin, and the cumulative dark current $\Phi\lambda_0$ can provide in excess of 50% of the signal energy, depending on the measurement (very noisy measurements). Design of $\Phi$ was based on Theorem 2, and $\lambda_0$ here is treated as the signature of an additional chemical (actually associated with measurement noise); finally, $\lambda = \Phi\lambda_0$ is the measurement dark current.

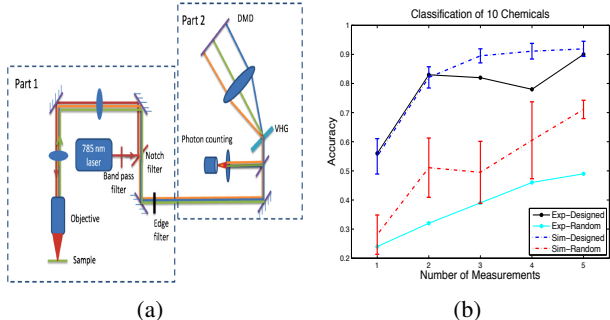

(a)  (b)

Figure 4: (a) Measurement system. The VHG is a volume holographic grating, that spatially spreads photons in a wavelength-dependent manner across the digital mirror microdevice (DMD), and the DMD is employed to implement binary coding. (b) Performance of the compressive-measurement classifier as a function of the number of compressive measurements; ten chemicals are considered. Experimental results are shown (Exp), as well as predictions from simulations (Sim).

The ten chemicals considered in this test were acetone, acetonitrile, benzene, dimethylacetamide, dioxane, ethanol, hexane, methylcyclohexane, octane, and toluene, and we note from Figure 4 that after only five compressive measurements excellent chemical classification is manifested based on designed CS measurements. There are $n > 1000$ wavelengths in a conventional measurement of these data, this system therefore reflecting significant compression. In Figure 4(b) we show results of measured data and performance predictions based on our model, with good agreement manifested. Note that designed projection measurements perform markedly better than random, where here the probability of a one in the random design was 10% (this yielded best random results in simulations).

## 5 Conclusions

New results are presented for the gradient of mutual information with respect to the measurement matrix and a dark current, within the context of a Poisson model for vector count data. The mutual information is considered for signal recovery and classification. For the former we recover known scalar results as a special case, and the latter results for classification have not been addressed in any form previously. Fundamental connections between the gradient of mutual information and conditional expectation estimates have been made for the Poisson model. Encouraging applications have been demonstrated for compressive topic modeling, and for compressive hyperspectral chemical sensing (with demonstration on a real compressive camera).

### Acknowledgments
The work reported here was supported in part by grants from ARO, DARPA, DOE, NGA and ONR.

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
