[Supplementary Material]

# Designed Measurements for Vector Count Data: Supplementary Material

[1]Liming Wang, [1]David Carlson, [2]Miguel Dias Rodrigues, [3]David Wilcox, [1]Robert Calderbank and [1]Lawrence Carin

[1]Department of Electrical and Computer Engineering, Duke University
[2]Department of Electronic and Electrical Engineering, University College London
[3]Department of Chemistry, Purdue University
{liming.w, david.carlson, robert.calderbank, lcarin}@duke.edu
m.rodrigues@ucl.ac.uk    wilcoxds@purdue.edu

## 1 Regularity Conditions

In this paper, we assume the following four regularity conditions (RC) on the interchangeability of integration and differentiation.

RC1:

$$\frac{\partial}{\partial \theta} \mathbb{E}_{Q_Y} \left[ f_{Y|X}^{\theta} \right] = \mathbb{E}_{Q_Y} \left[ \frac{\partial}{\partial \theta} f_{Y|X}^{\theta} \right], \tag{1}$$

RC2:

$$\frac{\partial}{\partial \theta} \mathbb{E}_{P_X} \left[ f_{Y|X}^{\theta} \right] = \mathbb{E}_{P_X} \left[ \frac{\partial}{\partial \theta} f_{Y|X}^{\theta} \right], \tag{2}$$

RC3:

$$\frac{\partial}{\partial \theta} \mathbb{E}_{P_X Q_Y} \left[ f_{Y|X}^{\theta} \log f_{Y|X}^{\theta} \right] = \mathbb{E}_{P_X Q_Y} \left[ \frac{\partial}{\partial \theta} \left( f_{Y|X}^{\theta} \log f_{Y|X}^{\theta} \right) \right]. \tag{3}$$

RC4:

$$\frac{\partial}{\partial \theta} \mathbb{E}_{Q_Y} \left[ f_{Y|X}^{\theta} \log f_{Y|X}^{\theta} \right] = \mathbb{E}_{Q_Y} \left[ \frac{\partial}{\partial \theta} \left( f_{Y|X}^{\theta} \log f_{Y|X}^{\theta} \right) \right]. \tag{4}$$

In addition, we always assume the technical condition that $\int \left[ \left| \log \frac{\mathrm{d}P_Y^{\theta}}{\mathrm{d}Q_Y} \frac{\partial}{\partial \theta} \left( \frac{\mathrm{d}P_{Y|X}^{\theta}}{\mathrm{d}Q_Y} \right) \right| \mathrm{d}P_X \mathrm{d}Q_Y \right] < \infty$.

## 2 Proof of Theorem 1

We first establish the following Lemma which relates to the results in [1].

**Lemma 1.** *Consider random variables $X \in \mathbb{R}^n$ and $Y \in \mathbb{R}^m$. Let $f_{Y|X}^{\theta}$ be the Radon-Nikodym derivative of the probability measure $P_{Y|X}^{\theta}$ with respect to arbitrary measures $Q_Y$ provided that $P_{Y|X}^{\theta} \ll Q_Y$. $\theta \in \mathbb{R}$ is a parameter. $f_Y^{\theta}$ is the Radon-Nikodym derivative of probability measure $P_Y^{\theta}$ with respect to $Q_Y$ provided that $P_Y^{\theta} \ll Q_Y$. Assume the regularity conditions RC1 – RC4, we have*

$$\frac{\partial}{\partial \theta} I(X; Y) = \mathbb{E} \left[ \frac{\partial \log f_{Y|X}^{\theta}}{\partial \theta} \log \frac{f_{Y|X}^{\theta}}{f_Y^{\theta}} \right]. \tag{5}$$

*Proof of Lemma 1.* Choose an arbitrary measure $Q_Y$ such that $P_{Y|X}^\theta \ll Q_Y$ and $P_Y^\theta \ll Q_Y$.

$$\frac{\partial}{\partial\theta}I(X;Y) = \frac{\partial}{\partial\theta}D(P_{Y|X}^\theta\|Q_Y) - D(P_Y^\theta\|Q_Y) \tag{6}$$

$$= \frac{\partial}{\partial\theta}\left[\int \log\frac{\mathrm{d}P_{Y|X}^\theta}{\mathrm{d}Q_Y}\frac{\mathrm{d}P_{Y|X}^\theta}{\mathrm{d}Q_Y}\mathrm{d}Q_Y\mathrm{d}P_X - \int \log\frac{\mathrm{d}P_Y^\theta}{\mathrm{d}Q_Y}\frac{\mathrm{d}P_Y^\theta}{\mathrm{d}Q_Y}\mathrm{d}Q_Y\right] \tag{7}$$

$$= \frac{\partial}{\partial\theta}\left[\int \log\frac{\mathrm{d}P_{Y|X}^\theta}{\mathrm{d}Q_Y}\mathrm{d}P_{Y|X}^\theta\mathrm{d}P_X - \int \log\frac{\mathrm{d}P_Y^\theta}{\mathrm{d}Q_Y}\frac{\mathrm{d}P_Y^\theta}{\mathrm{d}Q_Y}\mathrm{d}Q_Y\right]. \tag{8}$$

We will calculate the two terms in (8) separately.

$$\frac{\partial}{\partial\theta}\left[\int \log\frac{\mathrm{d}P_{Y|X}^\theta}{\mathrm{d}Q_Y}\mathrm{d}P_{Y|X}^\theta\mathrm{d}P_X\right] = \int\left[\frac{\partial}{\partial\theta}\left(\log\frac{\mathrm{d}P_{Y|X}^\theta}{\mathrm{d}Q_Y}\right)\mathrm{d}P_{Y|X}^\theta\mathrm{d}P_X\right]$$
$$+ \int\left[\log\frac{\mathrm{d}P_{Y|X}^\theta}{\mathrm{d}Q_Y}\frac{\partial}{\partial\theta}\left(\frac{\mathrm{d}P_{Y|X}^\theta}{\mathrm{d}Q_Y}\right)\mathrm{d}Q_Y\mathrm{d}P_X\right], \tag{9}$$

where the equality essentially follows from RC3. By Lemma 1 in [1], we have

$$\int\left[\frac{\partial}{\partial\theta}\left(\log\frac{\mathrm{d}P_{Y|X}^\theta}{\mathrm{d}Q_Y}\right)\mathrm{d}P_{Y|X}^\theta\mathrm{d}P_X\right] = 0. \tag{10}$$

Hence,

$$\frac{\partial}{\partial\theta}\left[\int \log\frac{\mathrm{d}P_{Y|X}^\theta}{\mathrm{d}Q_Y}\mathrm{d}P_{Y|X}^\theta\mathrm{d}P_X\right] = \int\left[\log\frac{\mathrm{d}P_{Y|X}^\theta}{\mathrm{d}Q_Y}\frac{\partial}{\partial\theta}\left(\frac{\mathrm{d}P_{Y|X}^\theta}{\mathrm{d}Q_Y}\right)\mathrm{d}Q_Y\mathrm{d}P_X\right] \tag{11}$$

$$= \int\left[\log\frac{\mathrm{d}P_{Y|X}^\theta}{\mathrm{d}Q_Y}\frac{\partial}{\partial\theta}\left(\log\frac{\mathrm{d}P_{Y|X}^\theta}{\mathrm{d}Q_Y}\right)\mathrm{d}P_{Y|X}^\theta\mathrm{d}P_X\right]. \tag{12}$$

The second term in (8) can be calculated as follow.

$$\frac{\partial}{\partial\theta}\left[\int \log\frac{\mathrm{d}P_Y^\theta}{\mathrm{d}Q_Y}\frac{\mathrm{d}P_Y^\theta}{\mathrm{d}Q_Y}\mathrm{d}Q_Y\right] \overset{RC4}{=} \int\left[\frac{\partial}{\partial\theta}\left(\log\frac{\mathrm{d}P_Y^\theta}{\mathrm{d}Q_Y}\right)\frac{\mathrm{d}P_Y^\theta}{\mathrm{d}Q_Y}\mathrm{d}Q_Y\right] + \int\left[\log\frac{\mathrm{d}P_Y^\theta}{\mathrm{d}Q_Y}\frac{\partial}{\partial\theta}\left(\frac{\mathrm{d}P_Y^\theta}{\mathrm{d}Q_Y}\right)\mathrm{d}Q_Y\right] \tag{13}$$

$$= \int\left[\frac{\partial}{\partial\theta}\left(\frac{\mathrm{d}P_Y^\theta}{\mathrm{d}Q_Y}\right)\mathrm{d}Q_Y\right] + \int\left[\log\frac{\mathrm{d}P_Y^\theta}{\mathrm{d}Q_Y}\frac{\partial}{\partial\theta}\left(\frac{\mathrm{d}P_Y^\theta}{\mathrm{d}Q_Y}\right)\mathrm{d}Q_Y\right] \tag{14}$$

$$\overset{RC1}{=} \frac{\partial}{\partial\theta}\int \mathrm{d}P_Y^\theta + \int\left[\log\frac{\mathrm{d}P_Y^\theta}{\mathrm{d}Q_Y}\frac{\partial}{\partial\theta}\left(\frac{\mathrm{d}P_Y^\theta}{\mathrm{d}Q_Y}\right)\mathrm{d}Q_Y\right] \tag{15}$$

$$= 0 + \int\left[\log\frac{\mathrm{d}P_Y^\theta}{\mathrm{d}Q_Y}\frac{\partial}{\partial\theta}\left(\int\frac{\mathrm{d}P_{Y|X}^\theta}{\mathrm{d}Q_Y}\mathrm{d}P_X\right)\mathrm{d}Q_Y\right] \tag{16}$$

$$\overset{RC2}{=} \int\left[\log\frac{\mathrm{d}P_Y^\theta}{\mathrm{d}Q_Y}\left(\int\frac{\partial}{\partial\theta}\left(\frac{\mathrm{d}P_{Y|X}^\theta}{\mathrm{d}Q_Y}\right)\mathrm{d}P_X\right)\mathrm{d}Q_Y\right] \tag{17}$$

$$= \int\left[\log\frac{\mathrm{d}P_Y^\theta}{\mathrm{d}Q_Y}\frac{\partial}{\partial\theta}\left(\frac{\mathrm{d}P_{Y|X}^\theta}{\mathrm{d}Q_Y}\right)\mathrm{d}P_X\mathrm{d}Q_Y\right] \tag{18}$$

$$= \int\left[\log\frac{\mathrm{d}P_Y^\theta}{\mathrm{d}Q_Y}\frac{\partial}{\partial\theta}\left(\log\frac{\mathrm{d}P_{Y|X}^\theta}{\mathrm{d}Q_Y}\right)\mathrm{d}P_{Y|X}^\theta\mathrm{d}P_X\right], \tag{19}$$

where the second to the last equality follows from the assumption together with the Fubini's theorem. We denote the specific regularity condition used on top of the corresponding equality symbol.

Plugging (12) and (19) back to (8), we have

$$\frac{\partial}{\partial\theta}I(X;Y) = \int\left[\log\frac{\mathrm{d}P^\theta_{Y|X}}{\mathrm{d}Q_Y}\frac{\partial}{\partial\theta}\left(\log\frac{\mathrm{d}P^\theta_{Y|X}}{\mathrm{d}Q_Y}\right)\mathrm{d}P^\theta_{Y|X}\mathrm{d}P_X\right] - \int\left[\log\frac{\mathrm{d}P^\theta_Y}{\mathrm{d}Q_Y}\frac{\partial}{\partial\theta}\left(\log\frac{\mathrm{d}P^\theta_{Y|X}}{\mathrm{d}Q_Y}\right)\mathrm{d}P^\theta_{Y|X}\mathrm{d}P_X\right]$$

(20)

$$= \int\left[\frac{\partial}{\partial\theta}\left(\log\frac{\mathrm{d}P^\theta_{Y|X}}{\mathrm{d}Q_Y}\right)\log\frac{\mathrm{d}P^\theta_{Y|X}/\mathrm{d}Q_Y}{\mathrm{d}P^\theta_Y/\mathrm{d}Q_Y}\mathrm{d}P^\theta_{Y|X}\mathrm{d}P_X\right]$$

(21)

$$= \mathbb{E}\left[\frac{\partial\log f^\theta_{Y|X}}{\partial\theta}\log\frac{f^\theta_{Y|X}}{f^\theta_Y}\right],$$

(22)

where the last equality follows from the definition of Radon-Nikodym derivatives $f^\theta_{Y|X}$ and $f^\theta_Y$. $\square$

*Proof of Theorem 1.* Let the parameter $\theta = \Phi_{ij}$. We first choose a measure $Q_Y$ such that $P^{\Phi_{ij}}_{Y|X} \ll Q_Y$ and $P^{\Phi_{ij}}_Y \ll Q_Y$. Let $f^{\Phi_{ij}}_{Y|X}$ and $f^{\Phi_{ij}}_Y$ be the Radon-Nikodym derivatives of $P^{\Phi_{ij}}_{Y|X}$ and $P^{\Phi_{ij}}_Y$, respectively. By Lemma 1, we have

$$\frac{\partial I(X;Y)}{\partial\Phi_{ij}} = \mathbb{E}\left(\frac{\partial}{\partial\Phi_{ij}}\log f^{\Phi_{ij}}_{Y|X}(Y|X)\times\log\frac{f^{\Phi_{ij}}_{Y|X}(Y|X)}{f^{\Phi_{ij}}_Y(Y)}\right)$$

(23)

$$= \mathbb{E}\left(\frac{\frac{\partial}{\partial\Phi_{ij}}f^{\Phi_{ij}}_{Y|X}(Y|X)}{f^{\Phi_{ij}}_{Y|X}(Y|X)}\times\log\frac{f^{\Phi_{ij}}_{Y|X}(Y|X)}{f^{\Phi_{ij}}_Y(Y)}\right).$$

(24)

Notice that by the Poisson channel assumption, $Y$ is supported on $\mathbb{Z}^m_+$. If we choose the measure $Q_Y$ to be the counting measure, then we have $f^{\Phi_{ij}}_{Y|X} = P^{\Phi_{ij}}_{Y|X}$ and $f^{\Phi_{ij}}_Y = P^{\Phi_{ij}}_Y$. Therefore, we have

$$\frac{\partial}{\partial\Phi_{ij}}f^{\Phi_{ij}}_{Y|X}(y|x) = \frac{\partial}{\partial\mathbf{M}_{ij}}\mathrm{Pois}(y;\Phi x+\lambda)$$

(25)

$$= \left(\frac{1}{y_i!}y_ix_j\times(\phi_ix+\lambda_i)^{y_i-1}\times e^{-(\phi_ix+\lambda_i)} + \frac{1}{y_i!}(\phi_ix+\lambda_i)^{y_i}(-x_j)e^{-(\phi_ix+\lambda_i)}\right)$$

$$\times\prod_{k\neq i}\frac{1}{y_k!}(\phi_kx+\lambda_k)^{y_k}e^{-(\phi_kx+\lambda_k)}$$

(26)

$$= \frac{1}{y_i!}x_j\times(\phi_ix+\lambda_i)^{y_i}e^{-(\phi_ix+\lambda_i)}\left(\frac{y_i}{\phi_ix+\lambda_i}-1\right)\times\prod_{k\neq i}\frac{1}{y_k!}(\phi_kx+\lambda_k)^{y_k}e^{-(\phi_kx+\lambda_k)}$$

(27)

$$= x_j\left(\frac{y_i}{\phi_ix+\lambda_i}-1\right)\times P^{\Phi_{ij}}_{Y|X}(y|x),$$

(28)

where $\phi_i$ is the $i$-th row of $\Phi$.

Therefore, we have

$$\frac{\partial I(X;Y)}{\partial\Phi_{ij}} = \mathbb{E}\left(X_j\left(\frac{Y_i}{\phi_iX+\lambda_i}-1\right)\times\log\frac{P^{\Phi_{ij}}_{Y|X}(Y|X)}{P^{\Phi_{ij}}_Y(Y)}\right)$$

(29)

$$= \mathbb{E}\left(X_j\left(\frac{Y_i}{\phi_iX+\lambda_i}-1\right)\times\log P^{\Phi_{ij}}_{Y|X}(Y|X)\right)$$

(30)

$$- \mathbb{E}\left(X_j\left(\frac{Y_i}{\phi_iX+\lambda_i}-1\right)\times\log P^{\Phi_{ij}}_Y(Y)\right).$$

(31)

We will calculate (30) and (31) separately. In the following derivations, we will omit the superscript $\Phi_{ij}$ in $P^{\Phi_{ij}}_{Y|X}(Y|X)$ and $P^{\Phi_{ij}}_Y(Y)$ for simplicity.

Term (30) may be expressed as

$$\mathbb{E}\left[X_j\left(\frac{Y_i}{\phi_i X + \lambda_i} - 1\right) \times \sum_k \log\left(\frac{1}{Y_k!}(\phi_i X + \lambda_i)^{Y_k} e^{-(\phi_i X + \lambda_i)}\right)\right] \tag{32}$$

$$= \sum_k \mathbb{E}\left[X_j\left(\frac{Y_i}{\phi_i X + \lambda_i} - 1\right)\log\frac{1}{Y_k!}\right] \tag{33}$$

$$+ \sum_k \mathbb{E}\left[X_j\left(\frac{Y_i}{\phi_i X + \lambda_i} - 1\right) \times Y_k \times \log(\phi_i X + \lambda_i)\right] \tag{34}$$

$$- \sum_k \mathbb{E}\left[X_j\left(\frac{Y_i}{\phi_i X + \lambda_i} - 1\right) \times (\phi_i X + \lambda_i)\right]. \tag{35}$$

We claim that (35) equals zero; this term may be expressed as

$$\sum_k \mathbb{E}\left[X_j\left(\frac{Y_i}{\phi_i X + \lambda_i} - 1\right) \times (\phi_i X + \lambda_i)\right]$$

$$= \sum_k \mathbb{E}\left[\mathbb{E}\left[X_j\left(\frac{Y_i}{\phi_i X + \lambda_i} - 1\right) \times (\phi_i X + \lambda_i)\bigg| X\right]\right] \tag{36}$$

$$= \sum_k \mathbb{E}\left[X_j\left(\frac{\mathbb{E}[Y_i|X]}{\phi_i X + \lambda_i} - 1\right) \times (\phi_i X + \lambda_i)\right] \tag{37}$$

$$= \sum_k \mathbb{E}\left[X_j\left(\frac{\phi_i X + \lambda_i}{\phi_i X + \lambda_i} - 1\right) \times (\phi_i X + \lambda_i)\right] \tag{38}$$

$$= 0, \tag{39}$$

where we use the fact that $\mathbb{E}[Y_i|X] = \mathbb{E}[\text{Pois}(Y_i; \phi_i X + \lambda_i)|X] = \phi_i X + \lambda_i$.

In turn, (30) may be expressed as

$$\sum_k \mathbb{E}\left[X_j\left(\frac{Y_i}{\phi_i X + \lambda_i} - 1\right)\log\frac{1}{Y_k!}\right] + \sum_k \mathbb{E}\left[X_j\left(\frac{Y_i}{\phi_i X + \lambda_i} - 1\right) \times Y_k \times \log(\phi_i X + \lambda_i)\right]. \tag{40}$$

Combining the fact that $\mathbb{E}[Y_i|X] = \phi_i X + \lambda_i$, $\mathbb{E}[Y_i^2|X] = (\phi_i X + \lambda_i) + (\phi_i X + \lambda_i)^2$ and $P_{Y|X}(y|x) = \prod_k P_{Y_k|X}(y_k|x)$, the latter term can be calculated as follow.

$$\sum_k \mathbb{E}\left[X_j\left(\frac{Y_i}{\phi_i X + \lambda_i} - 1\right) \times Y_k \times \log(\phi_i X + \lambda_i)\right]$$

$$= \sum_k \int x_j\left(\frac{y_i}{\phi_i x + \lambda_i} - 1\right) y_k \log(\phi_i x + \lambda_i)dP_X dP_{Y|X} \tag{41}$$

$$= \int x_j\left(\frac{y_i^2}{\phi_i x + \lambda_i} - y_i\right)\log(\phi_i x + \lambda_i)dP_X dP_{Y_i|X}$$

$$+ \sum_{k \neq i} \int x_j\left(\frac{y_i}{\phi_i x + \lambda_i} - 1\right) y_k \log(\phi_i x + \lambda_i)dP_X dP_{Y_i|X}dP_{Y_k|X} \tag{42}$$

$$= \int x_j\left(\frac{(\phi_i x + \lambda_i)^2 + (\phi_i x + \lambda_i)}{\phi_i x + \lambda_i} - \phi_i x + \lambda_i\right)\log(\phi_i x + \lambda_i)dP_X$$

$$+ \sum_{k \neq i} \int x_j\left(\frac{\phi_i x + \lambda_i}{\phi_i x + \lambda_i} - 1\right) y_k \log(\phi_i x + \lambda_i)dP_X dP_{Y_k|X} \tag{43}$$

$$= \mathbb{E}[X_j \log(\phi_i X + \lambda_i)] + 0. \tag{44}$$

We now establish the following technical Lemmas that will be used later. We note that the following Lemmas generalize the results in [1].

**Lemma 2.**

$$\mathbb{E}\left[\frac{X_j}{\phi_i X + \lambda_i}\Big| Y = y\right] = \frac{1}{y_i}\frac{P_Y(y_i - 1, y_i^c)}{P_Y(y)}. \tag{45}$$

*Proof of Lemma 2.* First observe that by the Poisson channel assumption, we have

$$\frac{1}{\phi_i x + \lambda_i} = \frac{1}{y_i}\frac{P_{Y_i|X}(y_i - 1|x)}{P_{Y_i|X}(y_i|x)} \tag{46}$$

$$\mathbb{E}\left[\frac{X_j}{\phi_i X + \lambda_i}\Big| Y = y\right] = \mathbb{E}\left[\frac{1}{y_i} \times \frac{P_{Y_i|X}(y_i - 1|X)}{P_{Y_i|X}(y_i|X)} \times X_j \Big| Y = y\right] \tag{47}$$

$$= \frac{1}{y_i}\int \frac{P_{Y_i|X}(y_i - 1|x)}{P_{Y_i|X}(y_i|x)} \times x_j dP_{X|Y=y} \tag{48}$$

$$= \frac{1}{y_i}\int \frac{P_{Y_i|X}(y_i - 1|x)}{P_{Y_i|X}(y_i|x)} \times x_j \frac{P_{Y|X}(y|x)}{P_Y(y)} dP_X \tag{49}$$

$$= \frac{1}{y_i}\frac{P_Y(y_i - 1, y_i^c)}{P_Y(y)}\int \frac{P_{Y_i|X}(y_i - 1|x)}{P_{Y_i|X}(y_i|x)P_Y(y_i - 1, y_i^c)}\prod_k P_{Y_k|X}(y_k|x) \times x_j dP_X \tag{50}$$

$$= \frac{1}{y_i}\frac{P_Y(y_i - 1, y_i^c)}{P_Y(y)}\int_x \frac{P_{Y_i|X}(y_i - 1|x)}{P_Y(y_i - 1, y_i^c)}\prod_{k\neq i} P_{Y_k|X}(y_k|x) \times x_j dP_X \tag{51}$$

$$= \frac{1}{y_i}\frac{P_Y(y_i - 1, y_i^c)}{P_Y(y)}\mathbb{E}[X_j|Y = (y_i - 1, y_i^c)]. \tag{52}$$

$\square$

**Lemma 3.**

$$\mathbb{E}(\phi_i X + \lambda_i|Y = y) = (y_i + 1)\frac{P_Y(y_i + 1, y_i^c)}{P_Y(y)}. \tag{53}$$

*Proof of Lemma 3.* First observe that

$$\phi_i x + \lambda_i = (y_i + 1)\frac{P_{Y_i|X}(y_i + 1|x)}{P_{Y_i|X}(y_i|x)}. \tag{54}$$

We have

$$\mathbb{E}(\phi_i X + \lambda_i|Y = y) = (y_i + 1)\mathbb{E}\left[\frac{P_{Y_i|X}(y_i + 1|X)}{P_{Y_i|X}(y_i|X)}\Big| Y = y\right] \tag{55}$$

$$= (y_i + 1)\int \frac{P_{Y_i|X}(y_i + 1|x)}{P_{Y_i|X}(y_i|x)} \times dP_{X|Y=y} \tag{56}$$

$$= \frac{y_i + 1}{P_Y(y)}\int_x \frac{P_{Y_i|X}(y_i + 1|x)}{P_{Y_i|X}(y_i|x)} \times P_{Y|X}(y|x) \times dP_X \tag{57}$$

$$= \frac{y_i + 1}{P_Y(y)}\int_x P_{Y_i|X}(y_i + 1|x)\prod_{k\neq i} P_{Y_k|X}(y_k|x) \times dP_X \tag{58}$$

$$= (y_i + 1)\frac{P_Y(y_i + 1, y_i^c)}{P_Y(y)}. \tag{59}$$

$\square$

**Lemma 4.**

$$\mathbb{E}\left[\frac{1}{\phi_i X + \lambda_i}\Big| Y = y\right] = \frac{1}{y_i}\frac{P_Y(y_i - 1, y_i^c)}{P_Y(y)}. \tag{60}$$

*Proof of Lemma 4.* From the same observation in the proof of Lemma 2, we have

$$\mathbb{E}\left[\left.\frac{1}{\phi_i X + \lambda_i}\right| Y = y\right] = \mathbb{E}\left[\left.\frac{1}{y_i} \times \frac{P_{Y_i|X}(y_i - 1|X)}{P_{Y_i|X}(y_i|X)}\right| Y = y\right] \tag{61}$$

$$= \frac{1}{y_i} \int \frac{P_{Y_i|X}(y_i - 1|x)}{P_{Y_i|X}(y_i|x)} dP_{X|Y=y} \tag{62}$$

$$= \frac{1}{y_i} \int \frac{P_{Y_i|X}(y_i - 1|x)}{P_{Y_i|X}(y_i|x)} \frac{P_{Y|X}(y|x)}{P_Y(y)} dP_X \tag{63}$$

$$= \frac{1}{y_i} \frac{1}{P_Y(y)} \int \frac{P_{Y_i|X}(y_i - 1|x)}{P_{Y_i|X}(y_i|x)} \prod_k P_{Y_k|X}(y_k|x) \times dP_X \tag{64}$$

$$= \frac{1}{y_i} \frac{1}{P_Y(y)} \int P_{Y_i|X}(y_i - 1|x) \prod_{k \neq i} P_{Y_k|X}(y_k|x) \times dP_X \tag{65}$$

$$= \frac{1}{y_i} \frac{P_Y(y_i - 1, y_i^c)}{P_Y(y)}. \tag{66}$$

$$\square$$

Combing previous derivations, we get

$$\frac{\partial I(X;Y)}{\partial \Phi_{ij}} = \mathbb{E}\left(X_j \log(\phi_i X + \lambda_i)\right) - \mathbb{E}\left\{X_j\left(\frac{Y_i}{\phi_i X + \lambda_i} - 1\right)\log\left(\left(\prod_k Y_k!\right)P_Y(Y)\right)\right\}$$

$$= \mathbb{E}\left(X_j \log(\phi_i X + \lambda_i)\right) - \mathbb{E}\left\{\left(\mathbb{E}\left(\frac{X_j}{\phi_i X + \lambda_i}\Big| Y\right)Y_i - \mathbb{E}(X_j|Y)\right)\log\left(\left(\prod_k Y_k!\right)P_Y(Y)\right)\right\} \tag{67}$$

$$= \mathbb{E}\left(X_j \log(\phi_i X + \lambda_i)\right)$$
$$- \mathbb{E}\left\{\frac{P_Y(Y_i - 1, Y_i^c)}{P_Y(Y)} \times \mathbb{E}\left[X_j|Y = (Y_i - 1, Y_i^c)\right] \times \log\left(\left(\prod_k Y_k!\right)P_Y(Y)\right)\right\}$$
$$+ \mathbb{E}\left\{(\mathbb{E}(X_j|Y))\log\left(\left(\prod_k Y_k!\right)P_Y(Y)\right)\right\} \tag{68}$$

$$= \mathbb{E}\left(X_j \log(\phi_i X + \lambda_i)\right)$$
$$- \int\left\{\frac{P_Y(y_i - 1, y_i^c)}{P_Y(y)} \times \mathbb{E}\left[X_j|Y = (y_i - 1, y_i^c)\right] \times \log\left(\left(\prod_k y_k!\right)P_Y(y)\right)\frac{dP_Y}{dQ_Y}dQ_Y\right\}$$
$$+ \mathbb{E}\left\{(\mathbb{E}(X_j|Y))\log\left(\left(\prod_k Y_k!\right)P_Y(Y)\right)\right\} \tag{69}$$

$$= \mathbb{E}\left(X_j \log(\phi_i X + \lambda_i)\right)$$
$$- \int\left\{P_Y(y) \times \mathbb{E}\left[X_j|Y = y\right] \times \log\left(\left((y_i + 1)!\prod_{k \neq i} y_k!\right)P_Y(y_i + 1, y_i^c)\right)dQ_Y\right\}$$
$$+ \mathbb{E}\left\{(\mathbb{E}(X_j|Y))\log\left(\left(\prod_k Y_k!\right)P_Y(Y)\right)\right\} \tag{70}$$

$$= \mathbb{E}\left(X_j \log(\phi_i X + \lambda_i)\right)$$
$$- \int\left\{\mathbb{E}\left[X_j|Y = y\right] \times \log\left(\left((y_i + 1)!\prod_{k \neq i} y_k!\right)P_Y(y_i + 1, y_i^c)\right)\frac{dP_Y}{dQ_Y}dQ_Y\right\}$$

$$+ \mathbb{E}\left\{ (\mathbb{E}(X_j|Y)) \log\left( (\prod_k Y_k!) p_Y(y) \right) \right\} \tag{71}$$

$$= \mathbb{E}\left( X_j \log(\phi_i X + \lambda_i) \right)$$

$$- \mathbb{E}\left\{ \mathbb{E}[X_j|Y] \times \log\left( ((Y_i+1)! \prod_{k\neq i} Y_k!) P_Y(Y_i+1, Y_i^c) \right) \right\}$$

$$+ \mathbb{E}\left\{ (\mathbb{E}(X_j|Y)) \log\left( (\prod_k Y_k!) P_Y(Y) \right) \right\} \tag{72}$$

$$= \mathbb{E}\left( X_j \log(\phi_i X + \lambda_i) \right) - \mathbb{E}\left\{ \mathbb{E}(X_j|Y) \log(Y_i+1) \frac{P_Y(Y_i+1, Y_i^c)}{P_Y(Y)} \right\} \tag{73}$$

$$= \mathbb{E}\left( X_j \log(\phi_i X + \lambda_i) \right) - \mathbb{E}[\mathbb{E}[X_j|Y] \log(\mathbb{E}[\phi_i X + \lambda_i|Y])], \tag{74}$$

where (68) follows from Lemma 2. (70) is obtained by a change of variable on $y_i$, together with the fact that $\frac{dP_Y}{dQ_Y} = P_Y$ for the counting measure $Q_Y$. (74) follows from Lemma 3.

Hence, we have

$$(\nabla_\Phi I(X;Y))_{ij} = \mathbb{E}\left[ X_j \log((\Phi X)_i + \lambda_i) \right] - \mathbb{E}\left[ \mathbb{E}[X_j|Y] \log \mathbb{E}[(\Phi X)_i + \lambda_i|Y] \right]. \tag{75}$$

Now we present the proof for the gradient of mutual information with respect to the dark current.

$$\frac{\partial I(X;Y)}{\partial \lambda_i} = \mathbb{E}\left( \frac{\partial}{\partial \lambda_i} \log f_{Y|X}^{\lambda_i}(y|x) \times \log \frac{f_{Y|X}^{\lambda_i}}{f_Y^{\lambda_i}} \right) \tag{76}$$

$$= \mathbb{E}\left( \frac{\frac{\partial}{\partial \lambda_i} f_{Y|X}^{\lambda_i}(y|x)}{f_{Y|X}^{\lambda_i}(y|x)} \times \log \frac{f_{Y|X}^{\lambda_i}}{f_Y^{\lambda_i}} \right). \tag{77}$$

Given the Poisson channel assumption, we can get that

$$\frac{\partial}{\partial \lambda_i} f_{Y|X}^{\lambda_i}(y|x) = \frac{\partial}{\partial \lambda_i} \mathrm{Pois}(y; \Phi x + \lambda) \tag{78}$$

$$= \left( \frac{1}{y_i!} y_i \times (\phi_i x + \lambda_i)^{y_i-1} \times e^{-(\phi_i x + \lambda_i)} + \frac{1}{y_i!}(\phi_i x + \lambda_i)^{y_i}(-e^{-(\phi_i x + \lambda_i)}) \right)$$

$$\times \prod_{k\neq i} \frac{1}{y_k!}(\phi_k x + \lambda_k)^{y_k} e^{-(\phi_k x + \lambda_k)} \tag{79}$$

$$= \frac{1}{y_i!} \times (\phi_i x + \lambda_i)^{y_i} e^{-(\phi_i x + \lambda_i)} \left( \frac{y_i}{\phi_i x + \lambda_i} - 1 \right) \times \prod_{k\neq i} \frac{1}{y_k!}(\phi_k x + \lambda_k)^{y_k} e^{-(\phi_k x + \lambda_k)} \tag{80}$$

$$= \left( \frac{y_i}{\phi_i x + \lambda_i} - 1 \right) \times P_{Y|X}^{\lambda_i}(y|x). \tag{81}$$

Followed by similar steps from (29) to (44), we obtain

$$\frac{\partial I(X;Y)}{\partial \lambda_i} = \mathbb{E}\left( \log(\phi_i X + \lambda_i) \right) - \mathbb{E}\left\{ \left( \frac{Y_i}{\phi_i X + \lambda_i} - 1 \right) \log\left( (\prod_k Y_k!) P_Y(Y) \right) \right\}$$

$$= \mathbb{E}\left( \log(\phi_i X + \lambda_i) \right) - \mathbb{E}\left\{ \left( \mathbb{E}\left( \frac{1}{\phi_i X + \lambda_i} \Big| Y \right) Y_i - 1 \right) \log\left( (\prod_k Y_k!) P_Y(Y) \right) \right\} \tag{82}$$

$$= \mathbb{E}\left( \log(\phi_i X + \lambda_i) \right)$$

$$- \mathbb{E}\left\{ \frac{P_Y(Y_i-1, Y_i^c)}{P_Y(Y)} \times \log\left( (\prod_k Y_k!) P_Y(Y) \right) \right\} + \mathbb{E}\left\{ \log\left( (\prod_k Y_k!) P_Y(Y) \right) \right\} \tag{83}$$

$$= \mathbb{E}\left(\log(\phi_i X + \lambda_i)\right)$$

$$- \int \left\{ \log\left( ((y_i+1)! \prod_{k \neq i} y_k!) P_Y(y_i+1, y_i^c) \right) dP_Y \right\} + \mathbb{E}\left\{ \log\left( (\prod_k Y_k!) P_Y(Y) \right) \right\} \tag{84}$$

$$= \mathbb{E}\left(\log(\phi_i X + \lambda_i)\right)$$

$$- \mathbb{E}\left\{ \log\left( ((Y_i+1)! \prod_{k \neq i} Y_k!) P_Y(Y_i+1, Y_i^c) \right) \right\} + \mathbb{E}\left\{ \log\left( (\prod_k Y_k!) P_Y(Y) \right) \right\} \tag{85}$$

$$= \mathbb{E}\left(\log(\phi_i X + \lambda_i)\right) - \mathbb{E}\left\{ \log(Y_i+1) \frac{P_Y(Y_i+1, Y_i^c)}{p_Y(Y)} \right\} \tag{86}$$

$$= \mathbb{E}\left(\log(\phi_i X + \lambda_i)\right) - \mathbb{E}\left[\log(\mathbb{E}[\phi_i X + \lambda_i | Y])\right], \tag{87}$$

where (83) and (87) follow from Lemma 4 and Lemma 3. (84) is obtained by a change of variable on $y_i$, together with the fact that $\frac{dP_Y}{dQ_Y} = P_Y$ for the counting measure $Q_Y$. Hence, we have

$$(\nabla_\lambda I(X;Y))_i = \mathbb{E}\left[\log((\Phi X)_i + \lambda_i)\right] - \mathbb{E}\left[\log \mathbb{E}[(\Phi X)_i + \lambda_i | Y]\right]. \tag{88}$$

$\square$

## 3   Proof of Theorem 2

*Proof.* First we notice that

$$I(C;Y) = H(Y) - H(Y|C) \tag{89}$$
$$= H(Y) - H(Y|X) + H(Y|X,C) - H(Y|C) \tag{90}$$
$$= I(X;Y) - I(X;Y|C), \tag{91}$$

where the second equality is due to the fact that $C \to X \to Y$ forms a Markov chain and $P_{Y|X,C} = P_{Y|X}$. Following by the similar steps in the proof of Theorem 1, we have

$$[\nabla_\Phi I(X;Y|C)]_{ij} = \left[ \mathbb{E}\left[X_j \log((\Phi X)_i + \lambda_i)\right] - \mathbb{E}\left[\mathbb{E}[X_j|Y,C] \log \mathbb{E}[(\Phi X)_i + \lambda_i | Y,C]\right] \right].$$

Hence,

$$[\nabla_\Phi I(C;Y)]_{ij} = -\mathbb{E}\left[\mathbb{E}[X_j|Y] \log \mathbb{E}[(\Phi X)_i + \lambda_i | Y]\right] + \mathbb{E}\left[\mathbb{E}[X_j|Y,C] \log \mathbb{E}[(\Phi X)_i + \lambda_i | Y,C]\right] \tag{92}$$

$$= -\mathbb{E}\left[\mathbb{E}\left[\mathbb{E}[X_j|Y,C] \middle| Y\right] \log \mathbb{E}[(\Phi X)_i + \lambda_i | Y]\right] + \mathbb{E}\left[\mathbb{E}[X_j|Y,C] \log \mathbb{E}[(\Phi X)_i + \lambda_i | Y,C]\right] \tag{93}$$

$$= -\mathbb{E}\left[\mathbb{E}\left[\mathbb{E}[X_j|Y,C]\right] \log \mathbb{E}[(\Phi X)_i + \lambda_i | Y] \middle| Y\right] + \mathbb{E}\left[\mathbb{E}[X_j|Y,C] \log \mathbb{E}[(\Phi X)_i + \lambda_i | Y,C]\right] \tag{94}$$

$$= -\mathbb{E}\left[\mathbb{E}[X_j|Y,C] \log \mathbb{E}[(\Phi X)_i + \lambda_i | Y]\right] + \mathbb{E}\left[\mathbb{E}[X_j|Y,C] \log \mathbb{E}[(\Phi X)_i + \lambda_i | Y,C]\right] \tag{95}$$

$$= \mathbb{E}\left[\mathbb{E}[X_j|Y,C] \log \frac{\mathbb{E}[(\Phi X)_i + \lambda_i | Y,C]}{\mathbb{E}[(\Phi X)_i + \lambda_i | Y]}\right] \tag{96}$$

Similarly, we have

$$(\nabla_\lambda I(X;Y|C))_i = \mathbb{E}\left[\log((\Phi X)_i + \lambda_i)\right] - \mathbb{E}\left[\log \mathbb{E}[(\Phi X)_i + \lambda_i | Y,C]\right]. \tag{97}$$

Therefore the gradient with respect to the dark current can be represented as

$$(\nabla_\lambda I(C;Y))_i = \mathbb{E}\left[\log \frac{\mathbb{E}[(\Phi X)_i + \lambda_i | Y,C]}{\mathbb{E}[(\Phi X)_i + \lambda_i | Y]}\right]. \tag{98}$$

$\square$

# 4  Variational Bayesian Updates for Topic Models

Given the model, $Y_d \sim \text{Pois}(\Psi S_d)$ with each column of $\Psi$, $\Psi_k$, drawn from a $\text{Dir}(\eta, ..., \eta)$ and each entry $S_{dk} \sim \text{Gamma}(\alpha_0, \beta_0)$. We let $Y_d \in \mathbb{Z}^n$, $S_d \in \mathbb{R}_+^K$ and $\Psi \in \mathbb{R}_+^{n \times K}$. We use Variational Bayesian updates to estimate the posterior distribution $q$:

$$\pi_{djk} \quad \propto \quad \exp\left(\psi(\gamma_{dk}) - \log(\beta_0 + 1) + \psi(\zeta_{kj}) - \psi(\sum_{i=1}^{n} \zeta_{ki})\right) \tag{99}$$

$$q(S_{dk}) \quad \sim \quad \text{Gamma}(\gamma_{dk}, \beta_0 + 1) \tag{100}$$

$$\gamma_{dk} \quad = \quad \alpha_0 + \sum_{j=1}^{n} Y_{dj} \pi_{djk} \tag{101}$$

$$q(\Psi_k) \quad \sim \quad \text{Dir}(\boldsymbol{\zeta}_k) \tag{102}$$

$$\zeta_{kj} \quad = \quad \eta + \sum_d Y_{dj} \pi_{djk} \tag{103}$$

where $\psi$ represents the digamma function.

When we consider the compressive measurements $Y_d | \Psi \sim \text{Pois}(\Phi \Psi S_d)$ where each $S_{dk} \sim \text{Gamma}(\alpha_0, \beta_0)$. In this case, we let $M = \Phi \Psi$ and $M \in \mathbb{R}_+^{m \times K}$, and we have $Y_d \in \mathbb{Z}^m$ and $S_d \in \mathbb{R}_+^K$. We use Variational Bayeisans to estimate the posterior distribution $q$:

$$\pi_{djk} \quad \propto \quad M_{jk} \exp\left(\psi(\gamma_{dk}) - \log(\beta'_{dk})\right)) \tag{104}$$

$$q(x_{dk}) \quad \sim \quad \text{Gamma}(\gamma_{dk}, \beta'_{dk}) \tag{105}$$

$$\gamma_{dk} \quad = \quad \alpha_0 + \sum_{j=1}^{n} Y_{dj} \pi_{djk} \tag{106}$$

$$\beta'_{dk} \quad = \quad \beta_0 + \sum_{j=1}^{m} M_{kj} \tag{107}$$

# References

[1] D.P. Palomar and S. Verdú. Representation of mutual information via input estimates. *IEEE Transactions on Information Theory,*, 53(2):453–470, Feb. 2007.