[Reviews · NeurIPS 2013]

Submitted by Assigned_Reviewer_4

This paper introduces an approach for optimally designing linear projections (Y) of (possibly noisy) data (X) that is well-modeled by a multidimensional Poisson distribution. The core new principle underlying this approach seems to be a closed-form expression of the gradient of the mutual information I(X; Y) with respect to the projection matrix. In classification-type situations where the data X arises from one of many classes C, the authors derive a similar expression of the gradient of the mutual information I(X; C). This closed-form gradient expression enables a simple descent-based optimization of the projection matrix. The authors provide evidence of the benefits of their proposed designed projection matrices via a number of experiments on document classification data as well as hyperspectral chemical sensor data.

The paper is well-written and organized, and it is refreshing to see the main ideas presented in such a coherent manner. The main theoretical development as well as the demonstrated results are very compelling. I have not checked the proofs in the supplementary material in detail, but they seem to make sense and agree with pre-existing (scalar) results.

There are a couple of points that the authors can perhaps clarify. First, the closed-form expressions for the gradients (mainly, Eqs 6 and 8) involve some complicated integrals that the authors propose to solve via Monte-Carlo sampling. It might be instructive to shed some more light on how much benefits this technique offers over (say) a brute-force computation of the gradient at each step. I do believe that the closed form expression eases a lot of computations but am unable to gauge exactly by how much.

Second, the application of their approach to document corpora is somewhat synthetic. It is unclear why a projection matrix is needed in this case when one typically has full access to the entire word-count vectors. The idea of "noise" in this context is also somewhat fuzzy (since the "projection" operator is being simulated in software). Moreover, since all calculations are being carried out in software anyway, it is unclear why the projection matrices are constrained to be binary. Why not use arbitrary real-valued projection matrices, especially if there is a chance to attain better performance? Regardless of these concerns, I agree that the experiments on the document datasets serve as a compelling proof-of-concept.
Summary: A very well-written paper with compelling results that proposes a new approach to optimally designing linear projection operators for Poisson-modeled data.

Submitted by Assigned_Reviewer_5

[Summary]
This paper studies the problem of designing linear measurements
for a vector Poisson signal model. The measurement matrix is designed
by maximizing mutual information between a vector of counts and a vector
Poisson rate or a latent class label. For this purpose the authors
derive the derivative of the mutual information with respect to
the measurement matrix and the dark current (Theorems 1 and 2),
which includes the known result on a scalar Poisson signal model
as a special case (Corollary 1). These results have been applied
to a problem of document characterization on the basis of word counts
and a problem of chemical sensing, demonstrating usefulness of
the proposed scheme for designing the measurement matrix.

[Quality]
This study is well motivated by a real-world application of chemical sensing
(although I feel the other problem on topic model rather artificial: I do not
see in what situations it would be easier to perform compressive measurements
on word counts, including extra efforts for designing optimal sets of words?),
and the theoretical arguments are directly connected to the application,
yielding theoretical results extending known results in the literature.
The experimental results show usefulness of the proposed approach. Overall,
this study has a nice balance between theory and application.

[Clarity]
This paper is basically clearly written, with a few errors.
I do not understand in the problem on the topic model how the author(s)
guaranteed the resulting word sets distinct, which is required in the
formulation as claimed in the line numbered 244.
In the line numbered 322, "(NNMF)" should be placed a bit earlier.
In the line numbered 348, S_{dk,p}' appearing just after 'where' would
read S_{dk,f}'.
In Figure 3 (b) and (c), there are only six columns but seven labels.
In the line numbered 352, there should be 'p' after "log".
In the line numbered 375, there should be 'p' after "argmax".
Some reference items lack essential bibliographic information,
such as volume and pages.

[Originality]
The subject of extending the Guo-Shamai-Verdu type identity to
a vector Poisson model is, as far as I know, original.

[Significance]
The work has theoretical importance, applying an information-theoretic
idea to design measurement matrices in compressive measurements on a
vector Poisson model. It also has practical significance in that it
would be applicable to a wide variety of applications.
Summary: Extension of an information-theoretic identity to a vector
Poisson model, which is in turn applied to a real-world problem of chemical
sensing. Of significance both theoretically and practically.

Submitted by Assigned_Reviewer_6

The authors consider Poisson measurement models for compressive measurements, which parallel the well-studied Gaussian measurement models. The main technical results are closed form expressions for the gradient of the mutual information between the input and output channel (with respect to the measurement matrix), and the gradient of the mutual information between the class label and output channel (also with respect to the measurement matrix). Apparently taking these information theoretic quantities into consideration when designing measurements is helpful (as opposed to simply choosing random measurement matrices).

The authors also provide some empirical results, though, at least for the newsgroup 20 dataset, the performance is rather poor (making it hard to know what one should glean as the take-home message). The theoretical results are also rather weak, and amount to a basic calculus calculation.
Summary: The main results are exercises in calculus, and the empirical results are not sufficiently compelling on their own to justify acceptance.

Submitted by Assigned_Reviewer_8

The paper proposes a design for measurement matrices of for compressed sensing for the case when the signal being measured is a Poisson vector. They propose selecting the design matrix to optimize for the mutual information I(X; Y) between the sensed signal (X) and the measurement (Y), which could be used for signal reconstruction. They also consider the design matrix chosen to optimize the mutual information I(Y; C) between the measurement and the underlying class (label) that generates the signal X. This could be used for classification purposes. They derive closed form expressions of I(Y; X) and I(Y; C), and use these to generate gradients. Their work is an extension of the work by Palomar/Verdu for the Gaussian case.

The main contribution theoretical contribution of the paper is the derivation of the gradient of the mutual information (which leads to an algorithm for selection of the design matrix). However, the quality of the resulting solution is not characterized.
The experimental results are also a little weak . The chemical sensing/classification problem, which seems to be the more interesting one, only uses 5 measurements - it would have been at least interesting to consider more measurements (at least for the simulation, because the graph shows the designed system to be saturating, while the accuracy of the randomly chosen matrix keeps increasing). Especially given that full reconstruction is often the goal, it would have been interesting to know the minimum number of measurements that are required to achieve the same quality as the full information case (both for the random and the designed matrices). The results from the 20 newsgroup dataset does not seem like a particularly good application either - the system as described is not better than the state of the art, and it is disappointing that the improvement obtained by using a model which looks like it takes significant amount of work to derive and compute generates roughly the same accruracy as one using twice the number of random measurements (in other words, using the vastly simpler random sensing matrix, which makes no assumption about the underlying model (and so is likely more robust), we need about 2x the number of measurements to achieve the same level of accuracy as a specifically designed model). The 2x factor could certainly be significant in a number of cases, but it is unclear if this is one of them. However, I think that this is a reasonable result to publish (perhaps simply to give people a better sense about the limits of designed sensing matrices). In). While the extensions to the Poisson model are not technically challenging, I found them interesting, especially the chemical sensing application. The connection between information theory and MMSE estimation is interesting. I dont think that the Palomar/Verdu result, connecting MMSE to the gradient of th emutaul information is a widely known result in the NIPS community (based on my unscientific poll of 5 people at least), and so its use and extension is likely a worthwhile addition to NIPS.


The paper is fairly clear (though many of the important details, such as the definition of the regularity conditions needed for their results has been shunted off to the supplementary material). The presentation of experimental results is a little sloppy though (the graphs are in color and are hard to read on a printed copy; the labels on the confusion matrix seem to be incorrect (the there are too many row labels in Figure 3(b) for example; it is pretty normal to use the same order in the rows and columns, which the authors dont do).
Summary: The paper considers an interesting problem (designing optimal sensing matrices), considers a restricted domain (vector poisson models, and cases where there is an underlying latent class structure) so that something theoretically interesting can be said about it, and derives an a gradient update equation for finding the optimal matrix. However, the theory feels incomplete because they only consider the gradients, but do not characterize the theoretical quality of the resulting matrix. The experimental results also feel incomplete, because they only consider a narrow range of measurement counts. Fixing up either one of this would make it a much more interesting read.
Author Feedback

Author rebuttal: Reviewer 1

The brute-force comparison was not performed because it would be very time-consuming to evaluate the high-D integral; the proposed Monte-Carlo sampling worked well in practice and could evaluate all of the gradients in under a second per step in our largest example. We agree that a greater degree of discussion on the computation issues is necessary and will be added.

We constrained the projection matrix to be binary because of its ease of implementation in real systems (such as imaging systems). Also, binary matrices for the document case have the pleasing (we feel) interpretation in terms of sets of words. However, we also implemented the system where the projection matrix was required to be positive-real with a total-energy constraint, and performance was similar to the binary-constrained matrix.

Concerning the document analysis, we don't dispute that this is somewhat artificial. However, for large vocabularies the projection measurements provide marked computational savings, with minimal degradation in performance. This could be of use in analysis of large corpora. Please note that there is a whole field dedicated to random projections of such matrices (we cited some of those papers), and we are the first to use info theory to design projections, which is much better than random.

Reviewer 2

Reviewer comment: I do not understand in the problem on the topic model how the author(s) guaranteed the resulting word sets distinct, which is required in the formulation as claimed in the line numbered 244.

Response: The word “distinct” will be changed to “different.” This wording is misleading--we allow the projections to choose the same word if it adds information to the system, although this rarely occurs.


Reviewer 3

Response to reviewer comments: In many real systems it is of interest to achieve the improved performance or decreased number of measurements. Fewer numbers of measurements allow greater throughput on devices—for example, in the chemical sensing problem we can decrease the number of measurements 3-fold. Since such systems are expensive, this is a significant real-world application that can have effects on the speed and results of scientific inquiry. The chemical system is a real and motivating example, and without CS the time required for measurements is prohibitive.

Reviewer comment: The authors also provide some empirical results, though, at least for the newsgroup 20 dataset, the performance is rather poor (making it hard to know what one should glean as the take-home message).

Response: The result on the 20 newsgroups was meant to show that it is possible to achieve near the same accuracy by using sets (compressive measurements) instead of analyzing each word individually. While this may not be done in practice, this is meant to show that by designing compressive measurements, one can obtain the similar performance. For the document analysis, particularly for large corpora, there is an opportunity for computational savings using the proposed approach.

Reviewer Comment: The theoretical results are also rather weak, and amount to a basic calculus calculation.

Response: The comment that this is a "basic calculus calculation" is, we feel, very inappropriate. We have significantly extended existing results, manifesting results for the vector Poisson case for the first time. References [1,12] were published in the IEEE T. Info Theory, which is a very competitive forum, and those results were only for the scalar case, and they had very limited if any real experiments. The extension of the gradient of the mutual information from the *scalar* Gaussian setting to the *scalar* Poisson setting is highly non-trivial (that is what [1,12] considered). We deal with an even more complex extension associated with the vector counterparts. This has never been done to date.

Further, we think it is fair to say that Shamai and Verdu are among the top ten living info theorists in the world (neither is an author of this paper). They would certainly not call our theory "just calculus," as their motivating work was published in IEEE T. Info Theory. So, while we can understand how someone not in this field may view our results as "just calculus," if not looked at carefully within the broader scope, what we have done is absolutely not just a calculus exercise.

Reviewer 4

Author Response: We will add additional simulation results on the chemical sensing so that the larger trends can be seen.

Reviewer Comment: The results from the 20 newsgroup dataset does not seem like a particularly good application either - the system as described is not better than the state of the art, and it is disappointing that the improvement obtained by using a model which looks like it takes significant amount of work to derive and compute generates roughly the same accuracy as one using twice the number of random measurements (in other words, using the vastly simpler random sensing matrix, which makes no assumption about the underlying model (and so is likely more robust), we need about 2x the number of measurements to achieve the same level of accuracy as a specifically designed model). The 2x factor could certainly be significant in a number of cases ... .

Response: The difference between our designed measurements and random measurements is much larger than 2x in many applications. In the 20 Newsgroups classification task, we get a 2x improvement over the heuristic techniques, such as NNMF or learning projections via an Latent Dirichlet Allocation (LDA) model; the NNMF takes comparable design time as our mutual information method and the LDA model takes longer to learn. We believe that some additional discussion of computational costs is necessary to show that our methods are actually quite computationally efficient (will be added). Designing a 100 by 8052 projection matrix took less than 1 minute via the gradient descent technique.